# Human Mitoribosome Biogenesis and Its Emerging Links to Disease

**DOI:** 10.3390/ijms22083827

**Published:** 2021-04-07

**Authors:** Maria Isabel G. Lopez Sanchez, Annika Krüger, Dmitrii I. Shiriaev, Yong Liu, Joanna Rorbach

**Affiliations:** 1Centre for Eye Research Australia, Royal Victorian Eye and Ear Hospital, 32 Gisborne Street, East Melbourne, VIC 3002, Australia; 2Department of Medical Biochemistry and Biophysics, Division of Molecular Metabolism, Karolinska Institutet, Solnavägen 9, 171 65 Stockholm, Sweden; annika.kruger@ki.se (A.K.); dmitrii.shiriaev@ki.se (D.I.S.); yong.liu@ki.se (Y.L.); 3Max Planck Institute Biology of Ageing—Karolinska Institutet Laboratory, Karolinska Institutet, 171 65 Stockholm, Sweden

**Keywords:** mitochondria, mitoribosome, assembly factors, rRNA, mitochondrial disease

## Abstract

Mammalian mitochondrial ribosomes (mitoribosomes) synthesize a small subset of proteins, which are essential components of the oxidative phosphorylation machinery. Therefore, their function is of fundamental importance to cellular metabolism. The assembly of mitoribosomes is a complex process that progresses through numerous maturation and protein-binding events coordinated by the actions of several assembly factors. Dysregulation of mitoribosome production is increasingly recognized as a contributor to metabolic and neurodegenerative diseases. In recent years, mutations in multiple components of the mitoribosome assembly machinery have been associated with a range of human pathologies, highlighting their importance to cell function and health. Here, we provide a review of our current understanding of mitoribosome biogenesis, highlighting the key factors involved in this process and the growing number of mutations in genes encoding mitoribosomal RNAs, proteins, and assembly factors that lead to human disease.

## 1. Introduction

Mammalian mitochondria contain their own ribosomes that synthesize 13 key subunits of the oxidative phosphorylation (OxPhos) system. While all RNA molecules required for mitochondrial translation, including 11 mitochondrial messenger RNAs (mt-mRNAs), 22 transfer RNAs (mt-tRNAs), and two ribosomal RNAs (rRNA), are encoded in the mitochondrial DNA (mtDNA), mitoribosomal proteins and auxiliary factors involved in translation are encoded in the nucleus.

Mitoribosomes differ significantly from their closest relatives, bacterial 70S ribosomes. This divergence brings unique features not only into the process of mitochondrial protein synthesis but also the mitoribosome biogenesis pathways. The process of assembly of prokaryotic and eukaryotic cytosolic ribosomes is relatively well-studied; however, many aspects of mitoribosome production remain poorly characterized. In this review, we summarize our current knowledge of mitoribosome biogenesis, focusing on the roles of assembly factors and provide their emerging links to disease.

### 1.1. Mammalian Mitoribosome

The mammalian mitoribosome consists of two subunits, a small 28S (mt-SSU) and a large 39S subunit (mt-LSU), which together form the 55S monosome. Full atomic models of the mt-LSU and mt-SSU at 3.4 to 3.8 Å were resolved concurrently by Ramakrishnan and Ban laboratories several years ago, providing unprecedented insight into the architecture of the mammalian mitoribosome [1,2,3,4]. The mt-SSU contains mtDNA-encoded 12S ribosomal rRNA and 30 nuclear DNA-encoded mitochondrial ribosomal proteins (MRPs), while the mt-LSU contains mtDNA-encoded 16S rRNA and 52 nuclear DNA-encoded MRPs. MRPs are mentioned in the text below using their new nomenclature suggested by [5] (e.g., MRPL1 is now uL1m). The 5S rRNA, located in the central protuberance (CP) of the bacterial LSU that is responsible for the interconnectivity between the large and the small subunit during translation, is absent in the mt-LSU. Instead, a mitochondrially-encoded tRNA (tRNA^Val^ or tRNA^Phe^) is found in the same position in mammals [1,2,6].

During evolution, the mammalian mitoribosome was subjected to a substantial reduction of RNA and increased protein mass. This led to a reversed RNA-protein ratio (~1:2) when compared with the bacterial ribosome (~2:1). There are 36 mitochondrial-specific proteins that create a network at the periphery of the structure, surrounding a conserved core. Moreover, many conserved homologs of bacterial ribosomal proteins acquired N- and C-terminal extensions. The newly recruited proteins and extensions do not directly compensate for the rRNA loss as they occupy different positions on the mitoribosome that do not match the missing rRNA segments, leading to a more porous overall structure.

Many regions of mitoribosomes underwent substantial rearrangements to adapt to the unique mitochondrial environment. The mitoribosomal tunnel, for example, is highly adapted for the synthesis of hydrophobic membrane proteins, with more hydrophobic walls compared with the bacterial tunnel [1,2]. Moreover, mitoribosome-specific protein mL45 is located in close proximity to the peptide tunnel exit to anchor mitoribosomes to the inner mitochondrial membrane. In this way, the nascent peptide can be co-translationally incorporated into OxPhos complexes by the Oxa1L translocase, which directly interacts with mL45 and uL24 [7]. As mitochondrial transcripts lack Shine–Dalgarno sequences [8], which in bacteria facilitate mRNA recognition by SSU, the mRNA entry site on the mt-SSU is largely remodeled in comparison to bacterial counterparts, with a mitochondria-specific pentatricopeptide repeat domain protein, mS39, serving as a platform for arriving leaderless mRNA [3,4]. Consequently, mitoribosomes exhibit a distinct translation initiation mechanism [9,10]. The unique structural features of the mitoribosome and the dual genomic origin of its components result in an assembly process that differs significantly from bacterial ribosome biogenesis.

### 1.2. Mitoribosome Biogenesis

During mitoribosome assembly, RNA components are transcribed, processed, and modified in mitochondria, while mitoribosomal proteins are imported from the cytosol. The essential steps required for mitoribosome maturation are schematically summarized in Figure 1.

mt-RNAs are transcribed as long polycistronic precursors in mtDNA-protein assemblies called nucleoids. The primary transcripts are processed through cleavage at the 5′ and 3′ ends by mitochondrial RNase P and ELAC2 (RNaseZ) nucleases, respectively, leading to the generation of individual mRNAs, tRNAs, and rRNAs [11,12]. Subsequently, various types of modifications, including methylation and pseudouridylation, contribute to rRNA maturation [13]. tRNA^Val^, which is integrated as a structural component of the human mitoribosome, undergoes additional modifications, including the addition of a CCA trinucleotide to the 3′-end and m^1^A9 modification [2,6].

Mitoribosome assembly is linked to the processing of mitochondrial transcripts, and likely begins co-transcriptionally, as the first stages of assembly were found to occur in RNA granules, near nucleoids, where RNA processing takes place [14]. While mitoribosomal proteins can form subcomplexes on unprocessed RNA, complete processing of mt-rRNAs is required for the successful assembly since the formation of mature mitoribosomal subunits is prevented by impaired 5′-end cleavage of the polycistronic transcript [15].

Important insights into the mechanism of mitoribosome assembly were revealed by pulse labeling with stable isotopes (SILAC), whereby incorporation of the ^13^C-labeled MRPs into the fully assembled mitoribosome was analyzed over time [16]. The resulting kinetic model of the mitoribosome assembly has been described in detail in [16] and is summarized in Figure 2. Interestingly, it was found that MRPs are generated and imported into mitochondria in excess, and non-incorporated proteins are consecutively degraded.

mt-SSU assembly likely starts with the folding of the 5′-end region of 12S rRNA, the first region emerging from mtRNA polymerase, most likely triggered by the early-binding clusters of MRPs (bS16m, mS40, mS27, mS34, and mS22). This region is subsequently bridged by uS5m with another early module that binds the 3′ region of 12S (uS7, uS9, mS29, mS31, mS35, and mS39). Other early MRPs then bind to the assembling particle as small clusters or individual proteins. Late-binding proteins also show a tendency to arrive in clusters, with a few remaining MRPs binding the growing mt-SSU separately, finalizing the assembly (Figure 2). Interestingly, SSU proteins classified as early-binding were previously reported to be highly abundant in nucleoids, further supporting a model whereby mitoribosome assembly is linked to transcription and processing [14]. Moreover, early MRPs mainly form the outer surface of the mt-SSU, whereas later proteins are more abundant at the intersubunit interface, which potentially prevents the association of premature mitoribosomes.

Assembly of the human mt-LSU is more complex as it requires the incorporation of a larger number of polypeptides and two RNA components: 16S and CP-tRNA^Val^. According to the kinetic model, 16S rRNA is first covered by four early clusters of MRPs and several individual proteins [16] (Figure 2). Incorporation of mL45 at the initial stages allows anchoring of the mt-LSU to the inner mitochondrial membrane. Thus, subsequent assembly steps occur in proximity to the membrane. A group of proteins (mL40, mL46, and mL48) that binds one facet of tRNA^Val^ in the central protuberance belongs to the early-binding group, while the other tRNA-binding group (mL38, uL18m, and bL27m) belongs to the intermediate assembly stage. When and how the tRNA component is incorporated and the role of assembly factors in this process remain to be elucidated.

Intermediate-binding proteins have scattered distribution on the mt-LSU, and their incorporation may depend on early-binding proteins [16]. A major cluster of intermediate-late proteins (uL23m, mL41, uL24m, uL29, and bL34m) joins the base of the mt-LSU to contribute to the formation of the highly hydrophobic peptide exit tunnel. The final steps of mt-LSU maturation include completion of the intersubunit surface and formation of the peptidyl transferase center (PTC), the ribosome’s catalytic core. Notably, the latter finding is consistent with the first structural cryogenic electron microscopy (cryo-EM) analysis of the mt-LSU late-stage assembly intermediates, where it was shown that folding of the PTC region is one of the final steps of 16S rRNA maturation, performed upon the binding of bL36m protein [17]. bL36m has not been identified in SILAC analyses [16], but the bacterial homolog of this protein is believed to participate in the PTC stabilization in late LSU assembly, further supporting this observation [19,20]. In conclusion, while the SILAC approach provided the first insights into the chronological order of MRP binding, a more detailed understanding of the process, especially around the involvement of the assembly factors at different stages of mitoribosome maturation, requires additional structural and biochemical investigations.

## 2. Mitoribosome Assembly Factors

Recent advances in structural biology, in particular cryo-EM, have given us a better understanding of the multi-step process of ribosome assembly. These studies revealed that many ribosome assembly factors function as macromolecular machines, which can increase efficiency and provide a higher level of control over this process. While several recent studies revealed extensive assembly machineries involved in mt-LSU and mt-SSU maturation in protozoa [21,22,23], so far, only one study on human mitoribosome late-stage assembly intermediates has been published [17]. Consequently, most of the knowledge summarized in the following section is derived from biochemical and cellular studies. Mitochondrial assembly factors can be classified according to their enzymatic functionalities, including guanosine triphosphate hydrolases (GTPases), helicases, pseudouridine synthases, methyltransferases, endonucleases, and factors without known enzymatic activity (here classified as other ribosomal assembly factors) (Figure 3). Many of the mitochondrial assembly factors identified so far are RNA-binding proteins, which associate with the 12S and 16S rRNAs, contributing to their correct folding.

### 2.1. GTPases

Guanosine triphosphate hydrolases (GTPases) encompass the largest group of assembly factors in bacteria and cytosolic ribosomes. So far, there are five GTPases known to function in mitoribosome assembly. Three of them were described to be involved in the maturation of the mt-LSU (GTPBP5, GTPBP10, and GTPBP7), and two in the maturation of the mt-SSU (ERAL1 and MTG3). Additionally, GTPBP8, a homolog of bacterial LSU assembly factor, is localized to mitochondria. All these GTPases were identified as components of the mitochondrial RNA granule proteome [30]. GTPases contain a highly conserved G-domain, which is capable of GTP hydrolysis and thus provides energy for the assembly process. This energy can be used for several processes, including the binding or removal of other proteins or the introduction of structural changes within assembling ribosomes. Furthermore, GTPases could act as reversible placeholders, which prevent binding to the premature ribosomal subunits, or as nutritional sensors, which function depends on cellular GTP/GDP levels [31]. Most GTPases involved in ribosome assembly contain an RNA-binding domain, which likely allows binding to rRNA. So far, there are no structural data available on human mitochondrial GTPase-ribosome complexes. Yet, a recent cryo-EM study on mitochondria from protozoa revealed that several GTPases can bind simultaneously to the assembling ribosome, allowing a coordinated mode of action [21]. As a very detailed review on the role of GTPases in mitochondrial ribosome biogenesis was published recently [32], we will only focus on the main aspects here.

#### 2.1.1. mt-SSU: ERAL1 and MTG3

There are two GTPases described so far that are involved in the maturation of the mt-SSU, ERAL1, and MTG3 (also known as NOA1 or C4ORF14). Loss of either of them results in mt-SSU assembly defects [33,34,35,36]. ERAL1 was suggested to be mainly involved in 12S rRNA maturation as it binds the hairpin at the 3′ terminus of 12S rRNA, where two highly conserved adenines undergo dimethylation by TFB1M [33]. Both the absence of ERAL1 as well as its accumulation result in decreased 12S rRNA levels and mt-SSU assembly defects [33,37]. Thus, correct ERAL1 protein levels seem to be critical for the assembly process [37], a mechanism that might also be essential for other assembly factors. Interestingly, interactome analysis of MTG3 identified ERAL1, suggesting an interplay between these two proteins [35].

#### 2.1.2. mt-LSU: GTPBP5 and GTPBP10

GTPBP5 (MTG2) and GTPBP10 are both homologous to the bacterial GTPase ObgE, which was shown to be involved in the late-stage maturation of bacterial LSU [38,39]. Interestingly, both proteins were shown to be able to rescue ObgE-depleted *E. coli* strains [40]. Yet, they seem to have independent functions in mitoribosome biogenesis and were suggested to work at different time points, with GTPBP10 binding to the maturing mt-LSU earlier than GTPBP5 [32]. Deciphering independent functions of these two GTPBPs is very challenging as they have several characteristics in common. Both proteins bind to the mt-LSU, especially to 16S rRNA, and depletion of either of them results in a significant reduction of monosome levels and attenuated mitochondrial translation [30,41,42,43]. Furthermore, interactome studies of both proteins showed high enrichment of MTERF4, NSUN4, and MALSU1 [41,42], which suggests the involvement of both proteins in late assembly stages. Interestingly, in contrast to GTPBP10, GTPBP5 showed high enrichment of MRM2 methyltransferase, which is in line with the observed hypomethylation of 16S rRNA in GTPBP5-depleted cells and suggests that GTPBP5 promotes MRM2 activity [41]. Furthermore, both proteins seem to have developed distinct functional mechanisms. While the presence of non-hydrolyzable GTP analog was shown to favor binding of GTPBP10 to the mt-LSU [43], the same GTP analog compromised GTPBP5 binding efficiency [44]. The exact functional mechanisms of both proteins still need to be elucidated.

#### 2.1.3. mt-LSU: GTPBP7 and GTPBP8

Based on the bacterial homologs of GTPBP7 (RbgA) and GTPBP8 (EngB), both GTPases likely bind to the rRNA of the mt-LSU and are involved in the maturation process of the PTC [45,46]. In line with this assumption, it was shown that GTPBP7 associates with the mt-LSU and 16S rRNA [47,48]. Notably, it was demonstrated that GTPBP7 additionally interacts with mS27 from the mt-SSU, a putative guanosine triphosphate exchange factor, and thus might be involved in intersubunit bridge formation. In contrast, besides the presence of GTPBP8 in mitochondrial RNA granules together with other GTPases [30,49], there are still no experimental data on the function of GTPBP8.

#### 2.1.4. mt-LSU/mt-SSU: GTPBP6

GTPBP6 shares a 30% sequence identity to bacterial HflX, which is a universally conserved protein and also a member of the Obg-GTPase family [50]. Structural studies showed that HflX is a heat shock-induced ribosome splitting factor binding to the PTC [51,52]. Interestingly, a recent study on human GTPBP6 demonstrated a potential dual function of this protein [53]. Besides facilitating the dissociation of mitoribosomes, GTPBP6 is involved in the mitoribosome assembly process. This assumption was based on the observation that deletion of GTPBP6 leads to the accumulation of late assembly intermediates of the mt-LSU containing MTERF4, NSUN4, MALSU1, GTPBP5, GTPBP7, and GTPBP10.

### 2.2. RNA Helicases

#### mt-LSU: DDX28 and DHX30

Ribosomal assembly requires the correct folding of the rRNA components. RNA helicases can function as RNA chaperone proteins and assist the necessary conformational changes. In a proteome analysis of mitochondrial RNA granules, two RNA helicases have been identified, which were suggested to participate in the assembly process of the mt-LSU [54]. This includes DDX28, which is a putative DEAD-box protein, and DHX30, which is a DEAH-box protein. Both protein families belong to the superfamily 2 RNA helicases but were shown to follow very distinct mechanisms [55]. While DDX28 was demonstrated to interact with the 16S during the early stages of mt-LSU assembly [49,54], the RNA substrate of DHX30 remains to be identified. Its involvement in mt-LSU biogenesis was suggested as depletion of DHX30 impaired mt-LSU maturation and monosome formation [54].

### 2.3. RNA Modifying Enzymes

Many rRNA modifications are found at or near functional sites within the ribosome, suggesting a regulatory role of rRNA modifications on ribosome function. To date, the existence of 10 modification sites in mitochondrial rRNA was demonstrated, including three 2′-O-ribose methylation sites, six base methylation sites, and one pseudouridylation site. Detection of RNA modifications is highly challenging; apart from their heterogeneity and potentially low abundance, dynamic changes can also hamper their identification [56]. It is, however, possible that more modification sites will be identified in the future due to the recent advances in experimental tools, such as sequencing techniques and cryo-EM. The use of the latter, for instance, allowed us to visualize all five methylation sites in 12S rRNA [10,57].

#### 2.3.1. RNA Methyltransferases

Methylation of mitoribosomal RNA is an emerging research field, and we are still at the beginning of understanding its biological significance. As this topic was covered in detail by a recent review [57], we will focus on the main aspects here.

##### mt-SSU: METTL15 and METTL17

Two recent studies on human cells identified METTL15 as the enzyme responsible for methylating position C839 in 12S rRNA [58,59]. Depletion of METTL15 resulted in impaired mitochondrial translation, demonstrating its significance in the assembly process. Interestingly, a study in mouse embryonic stem cells demonstrated decreased methylation at position C839 upon deletion of METTL17, another member of this protein family [60]. Additionally, decreased methylation at position C841 was observed, which was previously assigned to be methylated by NSUN4. Whether METTL17 plays a direct role in the methylation of those sites or the observed effects are caused by indirect pathways needs to be elucidated in future studies.

##### mt-SSU: TFB1M (and RBFA) and TRMT2B 

While early studies assigned TFB1M as a transcription factor analogous to TFB2M, consecutive studies demonstrated its role as a methyltransferase modifying residues A936 and A937 of 12S rRNA [61,62]. Recent crystal structures of a ternary complex of human TFB1M–12S(h45)–S-adenosyl-methionine and of a binary complex of TFB1M–12S(h45) provided the first insights into its mode of action [62]. *Tfb1m* knockout is embryonically lethal, and the loss of 12S rRNA dimethylation affects the stability of the mt-SSU in the murine model [63]. Notably, the dimethylation of A936 and A937 is not only dependent on the presence of TFB1M but also on the presence of RBFA, an RNA-binding protein that was shown to bind to this region of 12S rRNA [27]. Much less is known about the methyltransferase TRMT2B. While it was shown to modify residue U429 in 12S rRNA [64,65], the significance of this methylation is not yet clear, as the deletion of TRMT2B in a mouse model resulted in a very mild decrease in the activity of OxPhos complexes I, III and IV, but did not affect the mitochondrial translation rate [65].

##### mt-LSU/mt-SSU: NSUN4 (and MTERF4)

Increasing evidence suggests a dual function of NSUN4. Besides methylation of residue C841 in 12S rRNA [66], NSUN4 was shown to form a tight complex with MTERF4, which associates with the mt-LSU [24]. The essential role of both factors in the assembly process is apparent as the conditional lack of either factor in mice strongly decreases monosome formation and consequently mitochondrial translation [43,66]. Moreover, full-body knockouts of either factor resulted in embryonic lethality [43,66]. Structural insights were gained by two crystal structures of MTERF4-NSUN4 complexes [25,26]. Those studies demonstrated their strong interaction and furthermore revealed an RNA-binding path along both proteins, which potentially allows binding to 16S rRNA in the mt-LSU [25,26]. Whereas in vitro methylation assays demonstrated stimulation of 16S rRNA methylation of NSUN4 in the presence of MTERF4 [26], this has not been confirmed in vivo so far. Instead, analyses in mice suggest methylation activity of NSUN4 solely toward 12S rRNA that is independent of interaction with MTERF4 [66]. Consequently, the exact function of NSUN4 in complex with MTERF4 still needs to be clarified.

##### mt-LSU: MRM1, MRM2 and MRM3

Mitochondrial RNA methyltransferases (MRM) 1, 2, and 3 all perform methylation of ribose moieties in 16S rRNA (position G1145, U1369, and G1370, respectively) [67,68]. While methylation sites G1145 and U1369 are highly conserved among different species, methylation of residue G1370 is absent in bacteria. Still, it seems to be crucial for functional mitoribosomes as inactivation of MRM3, as well as MRM2, results in aberrant assembly of the mt-LSU and consequently loss of mitochondrial translation [68]. Additionally, it was demonstrated that both enzymes associate with the mt-LSU, suggesting that RNA methylation occurs directly at the mitoribosome, likely at the late-stage assembly [41,68]. A detailed characterization of MRM1, including the significance of its modification, is still not available.

##### mt-LSU: TRMT61B

While TRMT61B (tRNA (adenine[58]-N[1])-methyltransferase) was first described as a sole tRNA methyltransferase [69], it was later demonstrated to also mediate methylation of residue A947 in the mt-LSU, which is conserved in humans and all vertebrates [70]. The functional role of this modification, however, is still not explored.

#### 2.3.2. Pseudouridine Synthases

##### mt-LSU: RPUSD4 (Together with RPUSD3, TRUB2, NGRN, WBSCR16, and FASTKD2)

Pseudouridylation occurs by introducing a 180° rotation of the uracil base (i.e., the base becomes attached to ribose via its C5 atom). This changes the physicochemical properties of the nucleotide and consequently impacts RNA structure. Even though it is one of the most common RNA modifications, there is only one pseudouridylated site known in mt-rRNA. The enzyme responsible for introducing this modification is the pseudouridine synthase RPUSD4 [28]. This modification is important for mitoribosomal biogenesis, as demonstrated by the observed assembly defects of the mt-LSU and monosomes in *RPUSD4* knockout cells [28,71]. Interestingly, RPUSD4 was identified as part of a functional module including five other proteins, RPUSD3, TRUB2, NGRN, WBSCR16, and FASTKD2 [28,29]. Loss of any of these proteins was linked to decreased 16S rRNA levels and impaired mitochondrial translation [29]. Of note, even though all members of the protein module were shown to associate with 16S rRNA, the protein interactions were not dependent on the presence of RNA [29]. So far, only activity of RPUSD4 toward 16S rRNA could be demonstrated, and the mechanism of this multi-protein complex and its impact on pseudouridylation of 16S rRNA are still not clear. However, as there is strong evidence of a shared role [28,29], RPUSD3, TRUB2, NGRN, WBSCR16, and FASTKD2 are described together in this chapter even though they may not be directly involved in the pseudouridylation of rRNA.

RPUSD3, as well as TRUB2, are also mitochondrial pseudouridine synthases. Their enzymatic activities have so far only been demonstrated on mRNA and tRNA [28,72], and there is no evidence for their role in the pseudouridylation of rRNA. Still, both proteins were observed to co-sediment with mt-LSU, and depletion of either of them resulted in mt-LSU as well as monosome assembly defects [28]. NGRN lacks any common motif, and its function is largely unexplored. Yet, it was shown to be essential for functional OxPhos, and loss of NGRN triggers 16S rRNA depletion and consequently loss of mitochondrial translation [29]. WBSCR16 is a putative GDP/GTP exchange factor. A recent study analyzed three different splicing variants of WBSCR16, which are all localized exclusively to mitochondria [73]. However, only two of them are associated with mitoribosomes. Surprisingly, one splicing variant associates with the mt-LSU and the other one with the mt-SSU, suggesting independent functions of both splicing variants. In accordance with this observation, changes in their respective expression levels impacted the biogenesis of the mt-LSU and mt-SSU, respectively. As levels of the GTPases GTPBP10 and ERAL1 were also affected, they were suggested to be putative targets of WBSCR16 [73]. Besides its role in the ribosome assembly pathway, WBSCR16 was also shown to associate with OPA1 on the mitochondrial inner membrane and thus to be important for mitochondrial fusion [74]. The link between these different functions of WBSCR16 is still unknown. FASTKD2 is a member of the FAS-activated serine/threonine kinase (FASTK) family. Yet, the critical active site residues for kinase activity are not conserved within the protein family, and there is no evidence for the kinase activity of FASTKD2. All members are structurally related RNA-binding proteins containing three signature domains (FAST 1, FAST 2, and RAP) [75]. All localize to mitochondria but have distinct functions in the regulation of mitochondrial RNA biology [75]. Identified RNA targets of FASTKD2 are 16S rRNA and ND6 mRNA [76]. Accordingly, deletion of FASTKD2 resulted in reduced 16S rRNA levels and impaired assembly of the mt-LSU [54,76].

### 2.4. Endoribonucleases

#### mt-SSU: YBEY

YBEY is a homolog of the highly conserved YbeY proteins found in nearly all bacteria [77]. Bacterial YbeY is a strand-specific metallo-endoribonuclease, which is involved in late-stage assembly 70S ribosome quality control and in the maturation of the 5′ and 3′ ends of 16S rRNA as well as the 5′ termini of 23S and 5S rRNAs [78,79]. In human cells, it was shown to interact specifically with uS11m of the mt-SSU and thus was suggested to support the incorporation of uS11m into the maturating mt-SSU [80]. The involvement of YBEY in mitoribosome assembly was further supported by impaired mitochondrial translation and the respiratory deficiency caused by its deletion [80]. Even though the catalytic activity of recombinant human YBEY toward RNA was demonstrated, its substrate specificity and potential mechanism of action need further characterization.

### 2.5. Other Ribosomal Assembly Factors

#### 2.5.1. mt-LSU: MTERF3 and MTERF4

The MTERF protein family consists of four members, MTERF1, 2, 3, and 4. They share a common mTERF motif, which is based on repeats of a 30-residue module and contains leucine zipper-like heptads [81]. While MTERF1 and 2 are exclusively involved in mtDNA transcription and mtDNA replication, MTERF3 and 4 were suggested to participate in ribosome assembly. MTERF3 was first described as a negative regulator of mtDNA transcription initiation [82]. Later, it was demonstrated that decreased levels of MTERF3 not only activate mtDNA transcription but also result in reduced levels of 16S rRNA and impaired assembly of the mt-LSU [83]. Yet, it is not clear if this effect is due to the direct involvement of MTERF3 in ribosome assembly. In contrast, evidence for the involvement of MTERF4 in ribosome assembly is much stronger. It mainly acts in a complex with the methyltransferase NSUN4 and is therefore described in the methyltransferases section together with NSUN4.

#### 2.5.2. mt-LSU: MALSU1-LOR8F8-mt-ACP

The association of MALSU1 (mitochondrial assembly of ribosomal large subunit 1) with the mt-LSU and its essential role in the maturation process of the mt-LSU has been known for several years [84,85,86]. In 2017, the first high-resolution structure of a human mt-LSU assembly intermediate was solved by cryo-EM [17]. Interestingly, an additional density could be identified at the premature mt-LSU, which was assigned to MALSU1 and two additional proteins, LOR8F8 (LYR-motif-containing protein) and mt-ACP (acyl carrier protein). mt-ACP, also known as NDUFAB1, is a multifaceted protein with important functions in mitochondria [87]. Apart from being involved in mitoribosome biogenesis as part of the MALSU1–L0R8F8–mt-ACP module, it is involved in fatty acid synthesis, iron-sulfur cluster biogenesis, and is an integral component of respiratory complex I [88,89,90].

The structural analysis found the MALSU1–L0R8F8–mt-ACP module bound to uL14, and as it would sterically clash with the mt-SSU, it was suggested it prevents premature subunit joining. Notably, a recent cryo-EM structure of mt-LSU undergoing quality control identified the MALSU1 module together with the release factor C12orf65 and the RNA-binding protein C6orf203, indicating the involvement of the MALSU1 module not only in the assembly process but also in the recycling pathway of stalled mitoribosomes [7].

#### 2.5.3. mt-LSU: MPV17L2

MPV17L2 is a member of the MPV17 protein family, which comprises four integral membrane proteins. It is localized to the inner mitochondrial membrane and was demonstrated to interact with the mt-LSU as well as with the monosome [91]. As depletion of MPV17L2 results in decreased mt-LSU, mt-SSU, and monosome levels as well as impaired protein synthesis, it was suggested to play a role in the assembly process. Further studies are needed to reveal the exact function of MPV17L2 in the assembly process.

## 3. Post-Translational Modifications (PTMs) and Their Role in the Assembly Process

Recent improvements in mass spectrometry-based proteomics allowed the quantitative profiling of different types of post-translational modifications (PTMs). Considering the crucial role of PTMs in a wide range of biological processes, it is not surprising that many PTM maps identified ribosomal proteins as targets [92]. Observed PTMs include acetylation, phosphorylation, methylation, ubiquitination, and O-GlcNAcylation [93]. Increasing evidence suggests that ribosomal PTMs change upon different cellular stimuli, affecting protein-protein and protein-RNA interactions and consequently the assembly as well as the translation process (reviewed in [92,93]). However, the mechanistic aspects underlying those processes are still largely unknown. In mitochondria, several phosphorylation and acetylation sites have been mapped on the mitoribosome [94]. As most of them are localized to crucial structural regions within the mitoribosome and are conserved in bacteria, a regulatory function of those PTMs can be expected. Phosphorylation is a key regulatory element in mitochondria. Disturbances of phosphorylation patterns in mitochondria are closely connected to several disease phenotypes, including diabetes, cancer, and neurodegeneration (reviewed in [95]). Whether phosphorylation of mitochondrial-targeted proteins occurs before or after mitochondrial import, which kinases are involved respectively and how they are potentially imported into mitochondria still needs to be clarified. The first evidence for kinase activity on MRPs and the 55S ribosome was provided by a study from Koc et al. [96]. They observed translocation of the Fyn kinase, a Src family kinase, to mitochondria as well as its association with mitochondrial translation components. Furthermore, changes in its expression levels were linked to changes in mitochondrially-encoded complex IV subunits. Protein acetylation in mitochondria is thought to be mainly non-enzymatic, mediated by reactive lysine residues and acetyl-CoA. Even though an increasing number of site-specific acetylation sites have been identified in recent years, the functional consequences for most of them are still unknown (reviewed in [97]). In a proteomics approach, uL10m (MRPL10) was identified as the major acceptor of acetyl-groups within the mitochondrial ribosome [98]. Surprisingly, even though acetylation typically leads to loss of function, increased acetylation of uL10m was linked to increased translational activity. So far, to our knowledge, no study has addressed the overall significance of PTMs during ribosome biogenesis. Especially in mitochondria, the center of several metabolic pathways, strong crosstalk between nutrient availability and translational activity can be assumed, which potentially is regulated by PTMs. However, with improving proteomic approaches as well as high-resolution structures of mitoribosomes, this research field will certainly be explored in future studies.

## 4. Mitoribosome Biogenesis and Disease

In recent years, next-generation whole-exome sequencing has facilitated the discovery of disease-causing mutations in nuclear genes involved in different components of the mitochondrial protein translation machinery, including mitoribosome biogenesis and assembly. Pathogenic mutations have been identified in the RNA components of the mitoribosome (12S, 16S, and CP-tRNA^Val^), MRPs, and mitoribosome assembly factors. These mutations are associated with a wide range of clinical symptoms, variable tissue specificity, and can present at all stages of life. Notably, clinical progression is variable, and environmental or additional unknown genetic modifiers may alter disease severity and clinical outcome. In some cases, the underlying pathogenic mechanisms remain to be elucidated. Furthermore, numerous studies have shown links between the mitochondrial translation machinery, apoptotic signaling, and cancer susceptibility and progression [87]. Detailed information about mutations in different mitoribosome assembly components and other genetic-association studies relevant to disease are presented below and summarized in Table 1.

### 4.1. Pathogenic Variants in mt-rRNAs and CP-tRNA^Val^

#### 4.1.1. 12S rRNA

Aminoglycosides are wide-spectrum antibiotics commonly used against bacterial infections that work by blocking bacterial protein translation. However, some mtDNA mutations induce structural changes that make human mitoribosomes resemble bacterial ribosomes and enable a stronger aminoglycoside interaction [4]. Consequently, exposure to aminoglycosides can induce or worsen clinical symptoms in individuals with these mtDNA mutations. There are numerous mutations in the 12S rRNA, associated with non-syndromic and aminoglycoside-induced hearing loss (Figure 4 and Table 1), in addition to mutations in mt-tRNAs that are not incorporated into the mitoribosome [99]. While some of these mutations have been found in multiple individuals across different populations, others have only been found in single individuals and their pathogenic role in hearing loss requires further validation and characterization.

The mutations m.1555A > G and m.1494C > T have been found in multiple independent studies and are the best-characterized mtDNA mutations associated with aminoglycoside-induced and non-syndromic deafness [100,101,102]. Atypical clinical presentations of the m.1555A mutation have also been reported in individual pedigrees affected by hearing loss and cardiomyopathy or Leigh syndrome [103,104]. Both 1555 and 1494 bases are located in the highly conserved decoding region of the 12S rRNA in apposition to each other but do not form a base pair. However, the presence of either m.1555A > G or m.1494C > T induces the formation of a base pair between 1555A-1494T or 1494C-1555G, resulting in the addition of a terminal base pair at the end of the stem loop, making the mitoribosome structure similar to a bacterial ribosome at the aminoglycoside binding site. As a result, exposure to aminoglycosides can induce or worsen hearing loss in some individuals carrying these mutations, although additional risk factors may modulate the risk of ototoxicity [102].

#### 4.1.2. 16S rRNA

Four mutations in the 16S rRNA have been associated with disease, including atypical mitochondrial encephalopathy with lactic acidosis and stroke-like episodes syndrome (MELAS) (m.3093C > G, [105]), myopathy (m.3090G > A, [106]), cardiomyopathy (m.2336T > C, [107]), chronic progressive external ophthalmoplegia (CPEO), and Rett syndrome (2835T > C, [108] and [109]). The mutations m.3090G > A, m.3093C > G, and m.2835T > C are located in proximity to the PTC (Figure 4). The heteroplasmic m.3093C > G mutation was identified in a patient who also carried the well-characterized m.3243A > G MELAS-associated mutation, which may have contributed to the atypical MELAS presentation with diabetes mellitus, hyperthyroidism, and cardiomyopathy [105]. The mutation m.2336T > C was predicted to disturb the U2336-A2438 base pair in the stem-loop structure of the 16S rRNA, which is involved in mitoribosome assembly [107]. This pathogenic variant resulted in mitochondrial dysfunction and increased reactive oxygen species production in patient-derived lymphoblasts [107]. In addition to these mutations, high-resolution structural analysis of the human mitoribosome has enabled the discovery of rare, highly disruptive human variants in the 16S rRNA with the potential to cause disease [110].

#### 4.1.3. CP-tRNA^Val^

Multiple mutations in the gene *MT-TV*, encoding tRNA^Val^, have been associated with human disease. As tRNA^Val^ is not only a structural component of the mt-LSU but is also involved in amino acid delivery during the translation process, it is difficult to discriminate between the impact of mutations in this tRNA on mitoribosome assembly and translation elongation. Interestingly, it was shown that when the steady-state levels of tRNA^Val^ are reduced due to pathogenic mutation (m.1624C > T), tRNA^Phe^ is instead incorporated into the mt-LSU to prevent assembly defect [5]. This reveals remarkable plasticity of mitoribosome biogenesis; however, whether this is a common adaptive mechanism for all tRNA^Val^ mutations remains to be determined.

Clinical phenotypes of tRNA^Val^ mutations include MELAS (m.1642G > A; [111]), complex neurological presentations (m.1606G > A; [112] and m.1659T > C; [113]), MELAS and heart failure (m.1616A > G [114] and m.1644G > A [115]), Leigh syndrome (m.1624C > T; [116]), CPEO (m.1658T > C; [117]), and mitochondrial neurogastrointestinal encephalomyopathy (MNGIE; m.1630A > G; [118]). As with 12S rRNA mutations, substitutions in CP-tRNA^Val^ seem to be distributed across the whole molecule, targeting each structural part of the tRNA (Figure 4).

Notably, heteroplasmy levels, or the proportion of mutant and wild-type mtDNA, correlate with disease severity. This is best illustrated by the mutation m.1644G > A, which has significant variation in age at onset, symptoms, and disease progression. When present at high levels, this mutation has been associated with MELAS [119] and MELAS and hypertrophic cardiomyopathy [115]. However, a mutation load lower than 70% has been detected in asymptomatic patients [120]. G1644 is a highly conserved nucleotide, and the mutation alters the encoding part of the variable loop of the tRNA close to the base of the anticodon loop (Figure 4).

### 4.2. Pathogenic Variants in Mitoribosome Proteins

#### 4.2.1. mt-SSU MRPs

*bS1m* (in old nomenclature: *MRPS28*)—The c.356A > G (p.Lys119Arg) mutation in bS1m has been identified in multiple unrelated patients presenting with failure to thrive, sensorineural deafness, liver enlargement, facial dysmorphism, hypoglycemia, and lactic acidosis [121,122]. Patient fibroblasts displayed impaired mitoribosome biogenesis and mitochondrial translation defects [122]. bS1m is involved in the early stages of mitoribosome assembly [16] (Figure 2), and the c.356A > G mutation results in the substitution of a highly conserved lysine with arginine, which is predicted to destabilize protein, resulting in impaired mt-SSU biogenesis [122].

*uS2m (MRPS2)*—The biallelic mutations c.328C > T (p.Arg110Cys) and c.340G > A (p.Asp114Asn), and c.413G > A (p.Arg138His) in uS2m were identified in two unrelated patients respectively, presenting with sensorineural hearing impairment, developmental delay, hypoglycemia, and lactic acidemia [123]. These variants affect highly conserved amino acids and result in decreased mt-SSU abundance, while mt-LSU is unaffected, leading to inhibition of mitochondrial translation and multiple OxPhos deficiencies in patient fibroblasts [123]. This protein is involved in the early stages of mitoribosome assembly [16] (Figure 2) and forms most of its contacts with uS9m, mS23, and bS1m.

*bS6m (MRPS6)*—MRPS6 was one of 11 differentially expressed transcripts identified in a gene array on post-mortem brain samples from patients with Parkinson’s disease [124]. However, no protein validation or functional characterization was conducted, and therefore, the role of bS6m in Parkinson’s disease remains to be determined.

*uS7m (MRPS7)*—The mutation c.550A>G (p.Met184Val) in uS7m was associated with congenital sensorineural deafness, progressive hepatic and renal failure, and lactic acidemia in siblings [125]. uS7m is involved in the early stages of mitoribosome assembly [16] (Figure 2) and is located near the decoding center of the mt-SSU, sharing interactions predominantly with 12S rRNA, mS29 (DAP3), uS9m, and mS37. In silico analysis predicted that Met184 plays a critical role in stabilizing the folding of uS7m protein. Accordingly, the p.Met184Val mutation renders uS7m highly unstable and alters its RNA-binding ability. Decreased uS7m resulted in mitochondrial translation impairment and mitochondrial network fragmentation, which was attributed to reduced 12S rRNA transcript levels [125].

*uS9m (MRPS9)*—A case study reported a 360 kb deletion in 2q12.2q12.1 affecting only two genes, *POU3F3* and *MRPS9,* in a child presenting with intellectual disability and dysmorphic features [126]. Given the critical role POU3F3 plays in neuronal development [127], the clinical symptoms were attributed to the loss of this gene, although no functional characterization was conducted. Therefore, the role of *MRPS9* loss in disease remains unknown. The protein itself is localized at the outer surface of the subunit, where it forms extended contacts with 12S mt-rRNA and various MRPs, mostly with mS35, uS2m, uS7m, uS10m, and mS29 (DAP3).

*uS14m (MRPS14)*—The c.322C > T (p.Arg108Cys) mutation in uS14m was identified in one child affected by perinatal hypertrophic cardiomyopathy, neonatal lactic acidosis, growth delay, dysmorphic features, and neurological involvement [128]. Patient fibroblasts displayed mitochondrial translation defects. Notably, this is the first reported pathogenic MRP variant that does not affect mitoribosome assembly or stability, most likely due to uS14m binding to the mt-SSU during late-stage assembly. Instead, it is predicted to disrupt translation elongation or mt-mRNA recruitment [128].

*bS16m (MRPS16)*—The mutation c.331C > T (p.Arg111Ter) in bS16m was identified in one patient with congenital brain abnormalities, facial dysmorphism, brachydactyly, and metabolic acidosis resulting in perinatal death [129]. Patient fibroblasts had severely decreased mitochondrial translation, with reduced 12S rRNA transcript levels but unchanged 16S rRNA levels [129]. The protein is located in a pocket surrounded by 12S rRNA, mS34, mS40, mS22, mS25, and mS26.

*uS17m (MRPS17)*—A case study reported a 393 kb deletion in 7p11.2 affecting 11 genes, including *MRPS27*, in a child affected by psychomotor retardation and dysmorphic features [130]. Given the involvement of multiple genes and a lack of functional characterization, the role of *MRPS17* loss in this clinical case remains unknown. uS17m is localized at the outer surface of the subunit, where it forms contacts with 12S rRNA and several MPRs, including uS15m and mS25.

*mS18b (MRPS18-2)*—mS18b protein levels were found to be elevated in prostate cancer tissue; however, additional mechanistic characterization is required to establish a link between mS18b and cancer [131].

*mS22 (MRPS22)*—The mutation c.509G > A (p.Arg170His) in mS22 was identified in siblings presenting with edema, hypotonia, hypertrophic cardiomyopathy, and perinatal death [132]. Patient muscle biopsy showed decreased OxPhos enzymatic activity, significant reduction of 12S rRNA transcript, and moderate reduction of 16S transcript [132]. Additionally, the mutation c.644T > C (Leu215Pro) was identified in a child affected by Cornelia de Lange-like dysmorphic features, brain abnormalities, and hypertrophic cardiomyopathy [133]. In this case, patient fibroblasts had a generalized mitochondrial translation impairment and decreased 12S rRNA transcript levels, with 16S transcript levels mildly decreased [133]. These observations are consistent with the kinetic assembly model [15] (Figure 2), where mS22 belongs to the early assembly module, and consequently, its mutations may result in 12S rRNA instability.

Another clinical study described a homozygous, 4-bp duplication c.1032_1035dup, p.Leu346Asnfs*21 predicted to lead to a frameshift resulting in a change of the last 15 highly conserved amino acids of mS22 in a patient presenting with neonatal mitochondrial encephalopathy, congenital heart defects, and severe lactic acidosis [134]. In this case, reduced levels of mS22 were detected in patient fibroblasts, resulting in combined OxPhos deficiency. Finally, missense variants c.404G > A (p.Arg135Gln) and c.605G > A (p.Arg202His) have recently been identified in four females from two consanguineous families with primary ovarian insufficiency, although no effect on mS22 steady-state protein levels nor rRNA levels were observed in patient fibroblasts, implying a tissue-specific effect [135].

*mS23 (MRPS23)*—The mutation c.119C > G (p.Pro40Arg) in mS23 was identified in a single patient presenting with hepatic disease, with fibroblasts displaying decreased OxPhos complex I and IV enzymatic activity and reduced 12S rRNA abundance [136]. This early-binding protein is localized at the outer surface of mt-SSU and interacts with 12S rRNA and several MRPs, including uS2m, uS5m, and bS1m.

*mS25 (MRPS25)*—The mutation c.215C > T (p.P72L) in mS25 was identified in an individual presenting with encephalopathy, short stature, microcephaly, and dystonia [137]. Patient fibroblasts displayed a decreased steady-state level of mS25, indicating the mutation causes mS25 instability and reduced levels of other polypeptides of the mt-SSU, while mt-LSU proteins were unchanged. Although mS25 belongs to late-stage binding proteins [15] (Figure 2), the mitoribosome profile on sucrose gradient revealed very little intact mt-SSU, whereas the mt-LSU was normal, indicating that the mutation affects the assembly or stability of the mt-SSU [137]. This could be due to the disturbance of the mS25 close neighborhood, formed mostly by 12S rRNA, uS5m, bS16m, uS17m, mS22, and mS26.

*mS34 (MRPS34)*—Four pathogenic mutations in mS34 have been characterized in multiple unrelated individuals affected by Leigh-like or Leigh syndrome [138]. The homozygous splice-site mutations c.321 + 1G > T and c.322 − 10G > A resulted in abnormal mRNA splicing of the mS34 transcript and protein levels, while the compound heterozygous c.37G > A (p.Glu13Lys) missense variant and c.94C > T (p.Gln32*) nonsense variant led to decreased levels of mS34 protein only [138]. Fibroblasts or lymphoblasts from individuals with *MRPS34* mutations had combined OxPhos deficiencies and impaired mitochondrial translation attributed to decreased mt-SSU protein levels, leading to destabilization of the small mt-SSU and impaired monosome assembly [138]. The mutations may disrupt interactions of mS34 with 12S rRNA, as well as with several MRPs, such as mS40, mS26, and mS27.

*mS39 (MRPS39, PTCD3)*—The heterozygous variants c.415 − 2A>G and c.1747_1748insCT (p.Phe583Serfs*3) in mS39 were identified in a single individual with Leigh syndrome [139]. The mutations led to reduced mS39 transcript and protein levels and impaired mitochondrial translation. Patient fibroblasts showed a significant decrease in steady-state levels of mt-SSU proteins, while mt-LSU levels were unchanged, indicating that mt-LSU was still fully assembled despite the significant loss of the mt-SSU [139]. The mS39 is positioned at the mRNA entry tunnel, where it may facilitate the loading of the leaderless mRNA onto the monosome [18]. Its mutations may affect the binding of adjacent MRPs (mS35, uS3m, uS5m, uS10m, mS31, and mS33).

#### 4.2.2. mt-LSU MRPs

*uL3m (MRPL3)*—Mutations in uL3m have been associated with cardiomyopathy [140] and neonatal lactic acidosis, sensorineural hearing loss, cirrhosis, and interstitial nephritis [141]. Characterization of the c.950C > G (p.Pro317Arg) mutation in patient fibroblasts revealed that it leads to altered mitoribosome assembly, mitochondrial translation deficiency, and abnormal assembly of respiratory chain complexes [140]. Structural models of the mt-LSU suggest the mutations could alter the binding of uL3m, which is an early-binding protein located at the outer surface in the pocket formed by 16S rRNA, mL39, uL13m, bL17m, and bL19m.

*bL12m (MRPL12)*—A single mutation in bL12m (c.542C > T; p.Ala181Val) has been associated with clinical symptoms, including neonatal failure to thrive, muscle weakness, and abnormal neurological development with psychomotor symptoms [142]. The c.542C > T mutation affects a highly conserved alanine, resulting in decreased steady-state levels of bL12m protein, its altered integration into the mt-LSU, and multiple respiratory chain enzyme deficiency in patient fibroblasts [142]. bL12m is an integral part of the L7/L12 stalk, where six copies of bL12m *n*-terminal domain interact with uL10 and mL53 [18]. Although the Ala181 region is not resolved in deposited structures of mammalian mitoribosomes, in silico modeling of the region and its superimposition with bacterial ribosome structure indicated the p.Ala181Val mutation has a high probability of altering interactions of bL12m with elongation factors, which is an important function of the L7/L12 stalk [142].

*uL24m (MRPL24)*—The mutation c.272T > C (p.Leu91Pro) in uL24 was identified in a single patient affected by cerebellar atrophy, choreoathetosis, intellectual disability, and cardiac symptoms characteristic of Wolff-Parkinson-White syndrome [143]. Patient fibroblasts had reduced levels of assembled mt-LSU and decreased mitochondrial translation [143]. uL24m interacts with the 16S rRNA and other MRPs (uL23m, mL45, uL29m, and mL41) to contribute to the formation of the peptide exit tunnel [4]. The c.272T > C mutation is predicted to destabilize the structure of uL24m, leading to aberrant folding and altered interaction with other components of the mt-LSU, especially 16S rRNA and mL45 [143].

*mL44 (MRPL44)*—The c.467T > G (p.Leu156Arg) mutation in mL44 was identified in multiple unrelated patients presenting with childhood-onset hypertrophic cardiomyopathy and additional neurological and neuro-ophthalmological symptoms manifesting in adulthood [144,145]. One patient was found to have the compound heterozygous missense mutation c.233G  > A (p.Arg78Gln) in addition to c.467T > G in mL44 [145]. The c.467T>G mutation affects the assembly of the mt-LSU and COX1 stability but has no effect on de novo synthesis of mitochondrial polypeptides [143]. Notably, this mutation causes tissue-specific respiratory chain impairments, with complex I and IV deficiency in patient heart and skeletal muscle, but only isolated complex IV deficiency in fibroblasts [143]. Structural analysis suggests that the mutation could destabilize the protein and disrupt its interactions with 16S rRNA and neighboring MRPs, mostly with uL13m, mL42, and mL43.

### 4.3. Genetic Associations of Mitoribosome Assembly Factors with Disease

#### 4.3.1. GTPases

##### GTPBPs of mt-LSU Assembly Pathway

The loss of *GTPBP5* has been associated with a heterogeneous disorder characterized by congenital malformations. A 0.7 Mb deletion on chromosome 20q13.33 was identified in one pediatric patient with congenital malformations involving trachea-esophageal fistula, esophageal atresia, and cardiac anomalies [146]. Although the case report attributed the clinical phenotype to the loss of *GTPBP5*, the deletion resulted in the loss of 10 additional genes, and therefore, further characterization is required to determine the specific role of GTPBP5 in this context.

Gene expression analyses have found several associations between ribosomal GTPases and disease. Microarray analysis of samples from individuals with Klinefelter’s syndrome identified over-expression of *GTPBP6* to be inversely correlated with verbal ability and IQ [147]. Increased protein levels of GTPBP7, also known as MTG1, were detected in samples from patients with pathological cardiac hypertrophy [148]. Copy number variation analysis identified *GTPBP10*, among 11 additional genes, as having a high correlation with patient outcome in a clinically aggressive form of glioblastoma [149]. Similarly, expression levels of *GTPBP10* and five additional genes were found to have a strong association with prostate cancer biochemical recurrence [150]. However, future studies are required to determine the functional significance of GTPases in these contexts.

A homozygous missense mutation in *ERAL1* (c.707A > T; p.Asn236Ile) was identified in three unrelated patients with symptoms associated with Perrault syndrome, including sensorineural hearing loss, ovarian dysfunction, and infertility in females [151]. Patient fibroblasts showed decreased 12S rRNA and MRPS22 protein levels, suggesting a pathogenic role for the p.Asn236Ile variant.

#### 4.3.2. RNA Helicases

De novo missense variants in *DHX30* have been reported in multiple unrelated individuals presenting with developmental delay, intellectual disability, muscular hypotonia, and gait abnormalities [152,153]. In addition, a de novo likely pathogenic variant (c.2093C > T, p.Ser698Phe) was identified in infant brothers with motor and cognitive delay, congenital clasped thumbs, and unilateral undescended testicles [154]. Notably, none of the affected individuals presented typical clinical signs of mitochondriopathies. Instead, the clinical manifestations appear to reflect DHX30′s role in global translation control during development.

Gene expression analyses identified *DDX28*, among other genes, to be differentially expressed in colorectal tumor tissue and a potential association between expression levels of *DDX28* and development and prognosis of early-onset colorectal cancer [155,156].

#### 4.3.3. RNA Methyltransferases and Functional Partners

*MRM2*—The homozygous missense variant Chr7: 2274933 C  >  T (p.Gly189Arg) in *MRM2* was identified in a pediatric patient with MELAS [157]. Although patient fibroblasts did not show diminished levels of Um1369 in 16S rRNA or impaired mitochondrial translation, the equivalent yeast Mrm2p p.Gly259Arg variant resulted in diminished Um2791 (corresponding to human 16S Um1369), confirming the pathogenicity of the p.Gly189Arg mutation [157]. This example illustrates the tissue specificity of pathogenic mutations in mitoribosome biogenesis factors. Although Gly189 is not involved in S -Adenosyl methionine (SAM) binding or catalysis, structural modeling of the p.Gly189Arg mutation predicted a potential new interaction with Asp149, which in turn forms a bond with Arg75, with potential knock-on effects on protein structure and catalytic function [157]. Furthermore, gene expression analysis identified *MRM2* (referred to as *FTSJ2*) as a new oncogene that is amplified in non-small cell lung cancer [158]. Another study found *MRM2* as the target of microRNA (miRNA)-542-3p [159]. miRNA-542-3p is down-regulated in non-small cell lung cancer cells, and it was reported that it may exert tumor-suppressive functions by targeting and upregulating *MRM2* [159].

*TRMT61B*—Currently, there are no highly penetrant variants in *TRMT61B* known to cause disease in isolation. However, astrocytes of Alzheimer’s disease patients displayed altered TRMT61B transcript levels [160]. Furthermore, functional and expression quantitative trait loci analyses linked TRMT61B to estrogen receptor-negative breast cancer [161]. Further research is needed to clarify the potential role of TRMT61B in human disease.

*METTL15*—A genome-wide survey identified a novel variant in *METTL15* (rs10835310) associated with childhood obesity [162].

*TFB1M—TFB1M* was initially thought to be a modifier gene for hearing loss associated with the m.A1555G mutation [163]; however, this was later questioned as patients with the m.A1555G mutation had similar 12S rRNA methylation levels to controls [164]. A separate study identified a common variant of human *TFB1M* (rs950994) associated with impaired insulin response, elevated glucose levels, and increased risk of type 2 diabetes [165]. Similar observations were documented for a mouse model with beta cell-specific knockout of *Tfb1m* that resulted in lower insulin secretion, mitochondrial dysfunction, and eventual development of type 2 diabetes [166].

*RBFA*—Copy number variation analysis identified *RBFA* (referred to as *C18orf22*), among several other genes, as a risk gene for autism spectrum disorder [167].

*NSUN4*—A study found that m5C-related genes play an essential role in tumor progression in hepatocellular carcinoma and that high expression of *NSUN4* correlated with survival outcome in this type of cancer [168].

#### 4.3.4. Functional Partners of Pseudouridine Transferases

*NGRN*—Serial analysis of gene expression identified 16 previously uncharacterized differentially expressed genes, including *NGRN*, as potentially novel pancreatic cancer markers [169], while microarray analysis showed that *NGRN* was up-regulated in the spinal cord tissue of patients with sporadic amyotrophic lateral sclerosis [170]. Further studies are required to determine the functional relevance of NGRN in these contexts.

*FASTKD2*—Mutations in *FASTKD2* have been found in multiple unrelated patients affected by mitochondriopathy. Initially, a study identified the mutation c.1246C > T (p.Arg416X) in *FASTKD2* in two siblings affected by mitochondrial encephalopathy with an unusual clinical presentation [171]. A subsequent case study reported compound heterozygous mutations (p.Arg205X and p.Leu255Pro) in *FASTKD2* in one patient with late age onset autosomal recessive MELAS-like syndrome with optic atrophy [172]. Recently, three additional mutations in *FASTKD2*, c.808_809insTTTCAGTTTTG, homoplasmic c.868C > T, and heteroplasmic c.1859delT/c.868C > T were identified in patients with heterogeneous clinical symptoms, including mitochondrial encephalomyopathy and hypertrophic cardiomyopathy [173]. These compound mutations resulted in multiple OxPhos complex deficiencies in patient lymphocytes and primary muscle tissue [173].

Other studies have suggested additional links between FASTKD2 and disease. A study found an association between dysregulated FASTKD2 and poor prognosis of patients with pancreatic ductal adenocarcinoma, through FASTKD2’s role in promoting tumor growth and invasion, via upregulation of c-Myc expression [174]. Furthermore, RNA sequencing identified FASTKD2 as one of several mitochondria-related differentially expressed transcripts in astrocytes from patients with Alzheimer’s disease, although no further characterization or validation were conducted [160].

#### 4.3.5. Endoribonuclease

A case study reported a 0.56-Mb microduplication of 21q22.3 encompassing nine genes, including *YBEY*, in a baby with congenital heart disease [175]. Furthermore, a large-scale whole-exome association study identified novel genes for genetic predisposition to breast cancer susceptibility, including *YBEY* [176]. Similarly, a genome-wide analysis of germline copy number variants identified loss of function alterations in several genes, including *YBEY*, which may predispose to colorectal adenoma formation [177]. Additional studies are required to validate and characterize the specific role of YBEY in disease.

#### 4.3.6. Pathogenic Variants in Other Ribosomal Assembly Factors

*MTERF3*—*MTERF3* (also referred to as *MTERFD1*) gene amplification and high protein expression levels were present in various tumor tissues, which was correlated with a lower survival rate in patients. This was also observed in a study showing that high MTERF3 expression correlates with poor prognosis in glioma cancer patients [178]. Additionally, MTERF3 over-expression promoted human colorectal cancer cell proliferation and irradiation resistance [179].

*mt-ACP (NDUFAB1)*—A gene expression network analysis identified *NDUFAB1* as one of several genes predicted to contribute to the occurrence and development of Alzheimer’s disease [180], while a genome-wide association meta-analysis identified three genes, including *NDUFAB1*, that may confer susceptibility to anxiety disorders [181]. Further studies are necessary to replicate, validate, and characterize the role of mt-ACP in neurological disease. Additional studies in mice have indicated that mt-ACP may act as a cardio-protector by enhancing energetic mitochondrial metabolism, as cardiac-specific deletion of *Ndufab1* resulted in bioenergetic defects and increased reactive oxygen species production, cardiomyopathy, and sudden death [182]. Moreover, a potential role for mt-ACP in protecting against obesity and insulin resistance was recently suggested [183]. Given the important, interconnected roles mt-ACP plays in mitochondrial metabolism [87], additional studies are needed to elucidate the specific mechanisms linking genetic variants in *NDUFAB1* and disease.

## 5. Future Perspectives

Investigations on human mitoribosome biogenesis have increased exponentially in recent years. While these studies have confirmed that several features of the assembly pathways of the human mitoribosome and its bacterial counterpart are similar, multiple differences are beginning to emerge. Although many factors required for human mitoribosome biogenesis have now been uncovered, it is likely that future research will extend the inventory of mitoribosome assembly factors even further. Major challenges for the next years will be not only to identify novel factors but also to determine the precise roles of individual factors that have already been identified and to understand the complex regulatory networks modulating the assembly process.

The application of newly developed cutting-edge techniques greatly advances our knowledge of ribosome biogenesis. This includes proteomics-based methods, which have already been successfully applied in human mitoribosome assembly studies [16]. High-resolution cryo-EM is also likely to provide a vast amount of information on the temporal order of ribosome biogenesis, with the first snapshots of the human mitoribosome assembly intermediates already obtained [16]. Furthermore, CRISPR/Cas9 gene-editing tools have expedited the generation of knockout and knock-in (e.g., carrying specific pathogenic variants) cell and animal models to validate the functions of the genes implicated in mitoribosome production, greatly accelerating research development. In summary, investigating the mechanisms of the mitoribosome biogenesis greatly enhances our understanding of the molecular mechanisms underlying the mitoribosome-related human diseases and may provide the cues for the development of future treatment strategies.

**Table 1 ijms-22-03827-t001:** Mutations in mitoribosome structural components and assembly factors associated with human disease. The listed mutations are mapped on the mitoribosome structure (Figure 4), except those labeled with an asterisk (*), since the corresponding residues are not resolved in the structures used in the analysis.

Mutation	Mitoribosome Structure	Clinical Symptoms	References
m.669T > C m.735A > G * m.745A > G m.801A > G m.827A > G m.839A > G m.856A > G m.961T > C * m.961delTinsC * m.961insC(*n*) * m.1005T > C m.1027A > G m.1095TC m.1116A > G * m.1494C > T m.1192C > A m.1192C > T m.1291T > C m.1310C > T m.1331A > G m.1374A > G m.1452T > C m.1517A > C m.1537C > T	12S rRNA	Non-syndromic hearing loss; aminoglycoside-induced hearing loss; sensorineural hearing loss	[102,184,185,186,187,188,189,190,191,192,193]
m.1555A > G	12S rRNA	Aminoglycoside-induced deafness; Leigh syndrome; cardiomyopathy	[100,101,103,104]
m.2336T > C	16S rRNA	Hypertrophic cardiomyopathy	[107]
m.3090G > A	16S rRNA	Myopathy	[106]
m.3093C > G	16S rRNA	Atypical MELAS syndrome	[105]
m.2835C > T	16S rRNA	Rett syndrome; CPEO	[108,109]
m.1606G > A	tRNA^Val^	Ataxia, seizures, mental deterioration, myopathy, and hearing loss	[112]
m.1616A > G	tRNA^Val^	MELAS and cardiomyopathy	[114]
m.1624C > T	tRNA^Val^	Leigh syndrome	[116]
m.1630A > G	tRNA^Val^	MNGIE	[118]
m.1642G > A	tRNA^Val^	MELAS	[111]
m.1644G > A	tRNA^Val^	Adult-onset Leigh syndrome; MELAS; MELAS and hypertrophic cardiomyopathy	[115,119]
m.1658T > C	tRNA^Val^	CPEO	[117]
m.1659T > C	tRNA^Val^	Learning difficulties, hemiplegia, movement disorder	[113]
c.356A > G (p.Lys119Arg)	bS1m (MRPS28)	Failure to thrive, sensorineural deafness, liver enlargement, facial dysmorphism, hypoglycemia, and lactic acidosis	[121,122]
c.328C > T (p.Arg110Cys) and c.340G > A (p.Asp114Asn); c.413G > A (p.Arg138His)	uS2m (MRPS2)	Sensorineural hearing impairment, developmental delay, hypoglycemia, lactic acidemia	[123]
Not applicable—differential expression	bS6m (MRPS6)	Parkinson’s disease	[124]
c.550A > G (p.Met184Val)	uS7m (MRPS7)	Congenital sensorineural deafness, progressive hepatic and renal failure, and lactic acidemia	[125]
360-kb deletion in 2q12.2q12.1 resulting in loss of *POU3F3* and *MRPS9*	uS9m (MRPS9)	Intellectual disability and dysmorphic features	[126]
c.322C > T (p.Arg108Cys)	uS14m (MRPS14)	Perinatal hypertrophic cardiomyopathy, neonatal lactic acidosis, growth retardation, dysmorphic features, and neurological involvement	[128]
c.331C > T (p.Arg111Ter)	bS16m (MRPS16)	Congenital brain abnormalities, facial dysmorphism, brachydactyly, and fatal lactic acidosis	[129]
393 kb deletion in 7p11.2 affecting 11 genes, including *MRPS27*	uS17m (MRPS17)	Psychomotor retardation and dysmorphic features	[130]
Not applicable—differential expression	mS18b (MRPS18-2)	Prostate cancer	[131]
c.644T > C (p.Leu215Pro); c.509G > A (p.Arg170His); c.404G > A (p.Arg135Gln); c.605G > A (p.Arg202His); c.1032_1035dup (p.Leu346Asnfs*21) *	mS22 (MRPS22)	Cornelia de Lange-like dysmorphic features, brain abnormalities, hypertrophic cardiomyopathy; skin edema, hypotonia, hypertrophic cardiomyopathy, and perinatal death; primary ovarian insufficiency; neonatal lactic acidosis, brain and heart abnormalities, perinatal death	[132,133,135]
c.119C > G (p.Pro40Arg)	mS23 (MRPS23)	Hepatic disease	[136]
c.215C > T (p.Pro72Leu)	mS25 (MRPS25)	Encephalopathy, short stature, microcephaly, and dystonia	[137]
c.321 + 1G > T, c.322 − 10G > A; c.37G > A (p.Glu13Lys); c.94C > T (p.Gln32*)	mS34 (MRPS34)	Leigh-like or Leigh syndrome	[138]
c.415-2A>G, c.1747_1748insCT (p.Phe583Serfs*3)	mS39 (MRPS39, PTCD3)	Leigh syndrome	[139]
c.950C > G (p.Pro317Arg) and large-scale deletion; c.49delC * p.(Arg17Aspfs*57) *	uL3m (MRPL3)	Cardiomyopathy; neonatal lactic acidosis, sensorineural hearing loss, cirrhosis, and interstitial nephritis	[140,141]
c.542C > T* (p.Ala181Val) *	bL12m (MRPL12)	Neonatal failure to thrive, muscle weakness, abnormal neurological development with psychomotor symptoms	[142]
c.272T > C (p.Leu91Pro)	uL24m (MRPL24)	Cerebellar atrophy, choreoathetosis, intellectual disability, Wolff-Parkinson-White syndrome	[143]
c.467T > G (p.Leu156Arg); c.233G > A (p.Arg78Gln)	mL44 (MRPL44)	Infantile cardiomyopathy; adult-onset retinopathy, hemiplegic migraine, Leigh-like lesions, renal insufficiency, and hepatopathy	[144,145]
0.7 Mb deletion in 20q13.33	GTPBP5	Congenital malformations involving trachea-esophageal fistula, esophageal atresia, and cardiac anomalies	[146]
Not applicable—differential expression	GTPBP6	Verbal ability and IQ in Klinefelter’s syndrome	[147]
Not applicable—differential expression	GTPBP7	Cardiac hypertrophy	[148]
Not applicable—copy number variation; gene association	GTPBP10	Glioblastoma clinical outcome; prostate cancer recurrence	[149,150]
c.707A > T (p.Asn236Ile)	ERAL1	Perrault syndrome	[151]
c.1478G > A (p.Arg493His), c.1685A > G (p.His562Arg), c.2342G > A (p.Gly781Asp), c.2344C > T (p.Arg782Trp), c.2353C > T (p.Arg785Cys), c.2354G > A p.Arg785His; c.2093C > T, p.Ser698Phe	DHX30	Developmental delay, intellectual disability, muscular hypotonia, and gait abnormalities; motor and cognitive delay, congenital clasped thumbs, and unilateral undescended testicles	[152,153,154]
Not applicable—differential expression	DDX28	Development and prognosis of colorectal cancer	[155,156]
Chr7: 2274933 C > T (p.Gly189Arg); not applicable (differential gene expression)	MRM2	MELAS; non-small cell lung cancer	[157,158,159]
Not applicable—differential expression	TRMT61B	Alzheimer’s disease; breast cancer	[160,161]
Not applicable—gene association	METTL15	Childhood obesity	[162]
Not applicable—intron variant	TFB1M	Type 2 diabetes risk	[165]
Not applicable—copy number variation	RBFA	Autism spectrum disorder risk	[167]
Not applicable—differential expression	NSUN4	Hepatocellular carcinoma survival outcome	[168]
Not applicable—differential expression	NGRN	Pancreatic cancer marker; sporadic amyotrophic lateral sclerosis	[169]
c.1246C > T (p.Arg416X); p.Arg205X and p.Leu255Pro; c.808_809insTTTCAGTTTTG, homoplasmic c.868C > T and heteroplasmic c.1859delT/c.868C>T	FASTKD2	Mitochondrial encephalopathy; late age onset autosomal recessive MELAS-like syndrome with optic atrophy; mitochondrial encephalomyopathy and hypertrophic cardiomyopathy	[171,172,173]
Not applicable—gene association; differential expression	FASTKD2	Pancreatic ductal adenocarcinoma prognosis; Alzheimer’s disease	[160,174]
0.56-Mb microduplication of 21q22.3, including YBEY	YBEY	Congenital heart disease	[175]
Not applicable—gene association; copy number variant	YBEY	Breast cancer susceptibility; colorectal adenoma formation	[176,177]
Not applicable—gene association; copy number variant	MTERF3	Multiple cancers	[178,179]
Not applicable—gene association; copy number variant	mt-ACP	Alzheimer’s disease; susceptibility to anxiety disorders	[180,181]

## Figures and Tables

**Figure 1 ijms-22-03827-f001:**
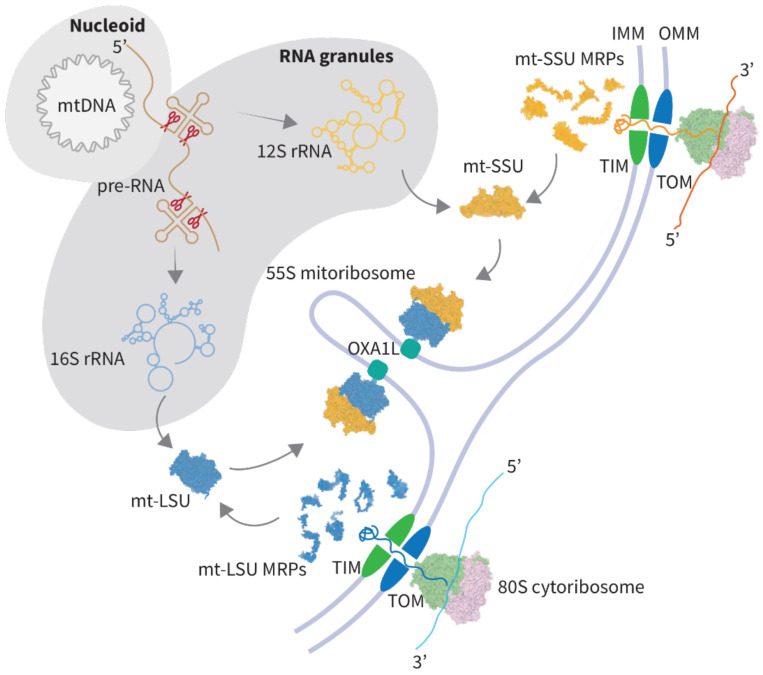
Biogenesis of the mitoribosome. The mitoribosome is composed of mitoribosomal proteins (MRPs) and RNAs, which are produced in the cytosol and mitochondria, respectively. The RNA components are derived from the transcription of mtDNA in nucleoids, followed by the RNA maturation in RNA granules. MRPs are synthesized by the 80S cytosolic ribosome and imported into the mitochondrial matrix through the mitochondrial import machinery (TOM and TIM proteins). Next, MRPs and rRNAs are assembled into the mitochondrial large subunit (mt-LSU) and the mitochondrial small subunit (mt-SSU). The complete 55S monosome is anchored to the inner mitochondrial membrane (IMM) and interacts with the translocase OXA1L. OMM-outer mitochondrial membrane.

**Figure 2 ijms-22-03827-f002:**
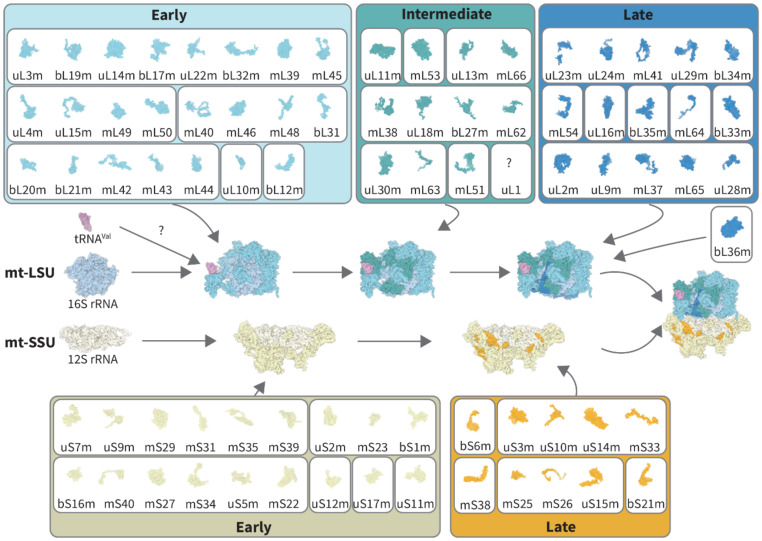
The assembly pathways of mt-LSU and mt-SSU. Components of the mt-LSU and mt-SSU are assembled in a hierarchical manner. The mt-LSU is assembled from 52 proteins, 16S rRNA, and mt-tRNA^Val^, whereas the mt-SSU is assembled from 30 proteins and 12S rRNA. Three main stages (early, intermediate, and late stages) have been identified in the mt-LSU, and two main stages (early and late stages) in the mt-SSU assembly pathways. Protein components integrating into the assembly intermediates in clusters are shown together, while individually recruited proteins are shown alone in boxes. Except for bL36m, the incorporation data are based on [16]. bL36m was integrated according to [17]. The timing of mt-tRNA^Val^ incorporation is still unknown (marked by a question mark), as well as the incorporation of mL52, mS37, and bS18m. All structures are based on the model PDB 6ZSG [18]. uL1m is not included in the figure due to its absence in 6ZSG; for uL12m, only the N-terminal structure of uL12m is depicted as it is the only part resolved in the 6ZSG model.

**Figure 3 ijms-22-03827-f003:**
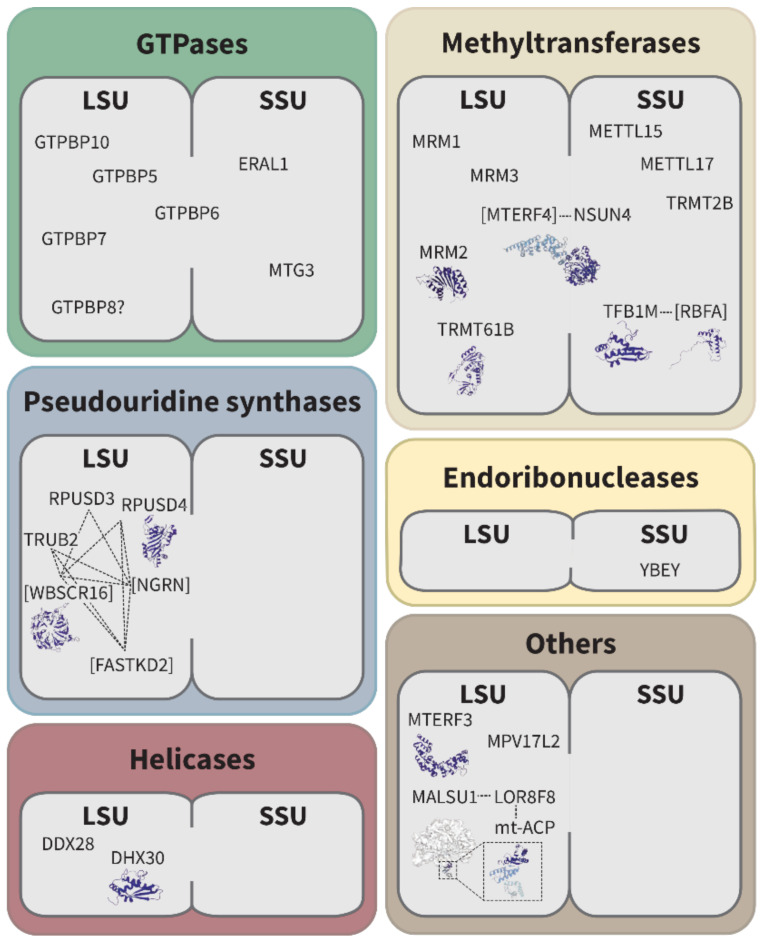
Mitoribosome assembly factors. Mitoribosomal assembly factors and their functional partners. Assembly factors without known enzymatic activity are termed “others”. Certain assembly factors are known to work in protein modules and are thus represented together and connected by dashed lines (NSUN4-MTERF4 [24,25,26]/TFB1M-RBFA [27]/RPUSD3-RPUSD4-TRUB2-WBSCR16-NGRN-FASTKD2 [28,29]). Factors of protein modules, which do not possess the corresponding enzymatic activity, are indicated in squared brackets. Even though RPUSD3 and TRUB2 are pseudouridine synthases, their activity toward 16S rRNA could not be demonstrated so far. GTPBP8 is speculated to participate in the assembly process due to its homology with the bacterial assembly factor. For all other proteins, some experimental evidence exists for their involvement in mitoribosome biogenesis. Available structures are depicted (RPUSD4 (pdb 5UBA), WBSCR16 (pdb 5XGS), DHX30 (pdb 2DB2), MRM2 (pdb 2NYU), TRMT61B (pdb 2B25), TFB1M (pdb 6AAX), RBFA (pdb 2E7G), NSUN4-MTERF4 (pdb 4FP9), and MTERF3 (pdb 3M66)). In the context of the mitoribosome, there is only structural data for the MALSU1 module available (MALSU1-LOR8F8-mt-ACP (PDB 500M)).

**Figure 4 ijms-22-03827-f004:**
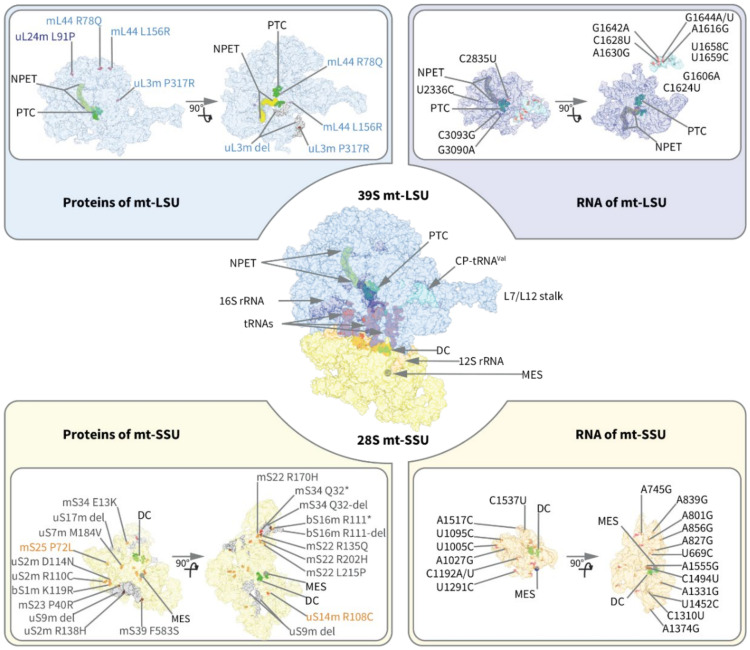
Disease-related mutations mapped on the structure of the human mitoribosome. Human mitochondrial ribosome (central panel). The mt-LSU contains mitoribosomal proteins (light blue), 16S mt-rRNA (blue), and CP-tRNA^Val^ (cyan), whereas the mt-SSU includes mitoribosomal proteins (yellow) and 12S mt-rRNA (orange). A-, P-, and E-site tRNA molecules (light red) bind both subunits of the mitoribosome. Key active sites of the mitoribosome: peptidyl transferase center (PTC) and decoding center (DC) are colored in green. Nascent peptide exit tunnel (NPET) is shown on the mt-LSU in yellow. mRNA entry site (MES) is shown on the mt-SSU in blue. Point mutations (red) or deletions (gray) may occur in both LSU (top) and SSU (bottom), targeting either protein components (left-hand side panel) or RNA (right-hand side panel). The colors of the protein names reflect the assembly stage they are incorporated into mitoribosome, according to the kinetic model (see Figure 2 for details). Each component (proteins or RNA) is shown in two orientations, the same as the original 55S monosome and rotated by 90° to depict the intersubunit surface. “del” indicates a complete deletion of a protein, while “X-del” notation indicates that residue X becomes a stop-codon and remaining C-terminal part is not present in the protein. Some of the mutations were not mapped since they are not resolved in the structure (see Table 1 for details).

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
