# Peer review of "Human Mitoribosome Biogenesis and Its Emerging Links to Disease"

_ijms, 2021, doi:10.3390/ijms22083827_

Round 1

Reviewer 1 Report

This is a well-organized, thorough, and thoughtful review on recent insights into the biogenesis of mitochondrial ribosomes and how this affects human health. The first portion of the manuscript concisely describes the assembly of both the small and large subunit of mitoribosomes, as well as describing a number of factors required for the assembly of these complex molecular machines. The second portion of the manuscript then examines known human mutations and, to a lesser extent, gene expression differences of mitoribosome subunits and assembly and how these relate to human disease. As the mitoribosome and its assembly factors comprise a large number of genes, this is not a trivial task. 

The authors have done a thorough job outlining the current literature. Further, the timing of this review is warranted, as the only reviews on the mitochondrial ribosome that I could find were published in 2016 or before. The current manuscript references over 70 papers published since 2017 (over one third of their citations), supporting the timeliness of an updated review in this field. 

I enjoyed reading this manuscript and feel it is suitable for publication with some minor edits and clarifications by the authors. My suggestions are as follows:

  1. In the first portion of the manuscript, the authors refer to mitochondrial ribosomal subunits by their now universal nomenclature, including the b/u/m annotations for conservation of these subunits. This nomenclature is not explained in the paper, and to make this manuscript more accessible to an audience that may not be familiar with this nomenclature, I suggest the authors define it at some point in their manuscript. Lines 45-46 may be a good spot (when the authors introduce the MRPs) or in the legend of Figure 2 would be my suggestions.
  2. In Figure 2, the individual proteins are shown in boxes that I believe indicate whether they assemble as a cluster or individually. The authors could clarify this by amending the figure legend to specify that proteins shown together in a box assembly in clusters whereas individually assembled proteins are shown alone in a box. (This is somewhat indicated in the current legend but not explicitly stated, which led to a bit of confusion for me as a reader. I think putting this small detail would clarify this well, as I think it is a very clever and informative way to graphically represent assembly intermediates.)
  3. In Figure 3, the authors do not specify what the lines represent in the figure - I assume these indicate physical interactions between these proteins, but I also do not know the source of the interactions. It would be helpful for the reader to clarify this.
  4. In the future perspectives portion of the manuscript, the authors note that CRISPR-Cas9 gene editing has expedited the generation of knockout cells and animals for studies. It may also be worth noting the capacity of CRISPR-Cas9 to generate models carrying specific pathogenic variants that could be used to explore the molecular mechanisms by which select mutations affect mitoribosome funciton. 
  5. Minor grammatical points: "synthesise" is misspelled in the abstract (line 13), there seems to be a word missing proximal to "mitoribosome" on line 219 - perhaps "the mitoribosome" or "mitoribosome biogenesis" would work, on line 378, the sentence starts "In human" which should be corrected to either "In humans" or "In human cells/patients/etc". 
  6. On an editorial note: if the manuscript as provided for review is formatted as it would be for final publication, it is hard for the reader to find lines 108-109 and 445-446 due to their placement over the figures on these pages.

Author Response

Response to Referee1:

We wish to thank the referee for dedicating their time to assess our manuscript and providing useful suggestions.

Below are our answers to the specific points raised by the reviewer:

  1. In the first portion of the manuscript, the authors refer to mitochondrial ribosomal subunits by their now universal nomenclature, including the b/u/m annotations for conservation of these subunits. This nomenclature is not explained in the paper, and to make this manuscript more accessible to an audience that may not be familiar with this nomenclature, I suggest the authors define it at some point in their manuscript. Lines 45-46 may be a good spot (when the authors introduce the MRPs) or in the legend of Figure 2 would be my suggestions.

Thank you for this important suggestion. We have now provided a short explanation of the new nomenclature: line 49: ‘MRPs are mentioned in the text below using their new nomenclature suggested by [PMID: 24524803] (e.g., MRPL1 is now uL1m)’; and in line 584: ‘bS1m (in old nomenclature: MRPS28)”.

  1. In Figure 2, the individual proteins are shown in boxes that I believe indicate whether they assemble as a cluster or individually. The authors could clarify this by amending the figure legend to specify that proteins shown together in a box assembly in clusters whereas individually assembled proteins are shown alone in a box. (This is somewhat indicated in the current legend but not explicitly stated, which led to a bit of confusion for me as a reader. I think putting this small detail would clarify this well, as I think it is a very clever and informative way to graphically represent assembly intermediates.)

We added additional information: Line 118: “Protein components integrating into the assembly intermediates in clusters are shown together, while individually assembled proteins are shown alone in boxes”.

  1. In Figure 3, the authors do not specify what the lines represent in the figure - I assume these indicate physical interactions between these proteins, but I also do not know the source of the interactions. It would be helpful for the reader to clarify this.

We thank the reviewer for spotting this lack of clarity. We added the corresponding references and explained the meaning of the dashed lines in the figure legend (Lines 182-184).

  1. In the future perspectives portion of the manuscript, the authors note that CRISPR-Cas9 gene editing has expedited the generation of knockout cells and animals for studies. It may also be worth noting the capacity of CRISPR-Cas9 to generate models carrying specific pathogenic variants that could be used to explore the molecular mechanisms by which select mutations affect mitoribosome funciton. 

We have changed line 873: “ knock-out and knock-in (e.g., carrying specific pathogenic variants) cell and animal models”.

  1. Minor grammatical points: "synthesise" is misspelled in the abstract (line 13), there seems to be a word missing proximal to "mitoribosome" on line 219 - perhaps "the mitoribosome" or "mitoribosome biogenesis" would work, on line 378, the sentence starts "In human" which should be corrected to either "In humans" or "In human cells/patients/etc". 

We have now changed ‘synthesise’ to ‘synthesize’, ‘mitoribosome’ to ‘mitoribosome biogenesis’, and ‘In human’ to ‘In human cells’.

  1. On an editorial note: if the manuscript as provided for review is formatted as it would be for final publication, it is hard for the reader to find lines 108-109 and 445-446 due to their placement over the figures on these pages.

Reviewer 2 Report

The review from Lopez Sanchez et al is an excellent review which aims to provide an update of the current knowledge we have regarding mitochondrial ribosome biogenesis. Besides, authors described the mitoribosome assembly factors and related mutations and genetic defects which have been found in patients. I have not found major issues while reading this manuscript. The figures are also very illustrative and offer good support for the reader.   

As a very minor point, I would suggest to include a short sentence indicating that mt-ACP (NDUFAB1) is a complex I subunit. In fact, authors correctly cited a couple of papers which explored the role of this protein not only as part of complex I but also as an important player during FeS-clusters biogenesis and assembly of other mitochondrial proteins. Thus, it would be helpful for the reader to relate this dual function of NDUFAB1.

I should say that this version of this manuscript is of high-quality and certainly ready to be accepted.

Author Response

We wish to thank the referee for their time to assess our manuscript and their positive comments. We have addressed the reviewer’s suggestion by adding the following (line 423-): 

mt-ACP, also known as NDUFAB1, is a multifaceted protein with important functions in mitochondria (PMID: 31473256). Apart from being involved in mitoribosome biogenesis as part of the MALSU1–L0R8F8–mt-ACP module, it is involved in fatty acid synthesis, iron-sulfur cluster biogenesis, and is an integral component of as a subunit of respiratory complex I (PMID: 27540631; PMID: 30118679; PMID: 25209663)” 

PMID: 31473256 Masud, A.J., Kastaniotis, A.J., Rahman, M.T., Autio, K.J., and Hiltunen, J.K. (2019). Mitochondrial acyl carrier protein (ACP) at the interface of metabolic state sensing and mitochondrial function. Biochim. Biophys. Acta - Mol. Cell Res. 1866, 118540. 

PMID: 27540631 Van Vranken, J.G., Jeong, M.-Y., Wei, P., Chen, Y.-C., Gygi, S.P., Winge, D.R., and Rutter, J. (2016). The mitochondrial acyl carrier protein (ACP) coordinates mitochondrial fatty acid synthesis with iron sulfur cluster biogenesis. Elife 5. 

PMID: 30118679 Van Vranken, J.G., Nowinski, S.M., Clowers, K.J., Jeong, M.-Y., Ouyang, Y., Berg, J.A., Gygi, J.P., Gygi, S.P., Winge, D.R., and Rutter, J. (2018). ACP Acylation Is an Acetyl-CoA-Dependent Modification Required for Electron Transport Chain Assembly. Mol. Cell 71, 567-580.e4. 

PMID: 25209663 Vinothkumar KR, Zhu J, Hirst J. Architecture of mammalian respiratory complex I. Nature. 2014 Nov 6;515(7525):80-84. doi: 10.1038/nature13686 

Reviewer 3 Report

This important manuscript by Lopez Sanchez et al. reviews the current state of matters in the biogenesis of mitochondrial ribosomes and highlights the vivid and multidimensional link between the impairment of mitoribosomal assembly and human diseases. It provides one of the most comprehensive and updated compendia of the mitoribosome subunits, RNA species, and factors that mediate their assembly to the functional ribosomes, together with description and consequences of the disease-associated variants. Several previous reviews described the assembly factors and structural features of the mitoribosomes. However, this paper's novelty stands on the attempt to bridge the structural and genetic data with the recently described kinetics of mitoribosome assembly. The figures are very clear and informative. The entire paper has been put together very nicely by the experts in the field and deserves to be published.

A few minor aspects might be considered:

  1. Subchapter 2.3. is very complex, and it would highly benefit from the additional figure that could help navigate through the text. For example, schematic representation of the modified areas of particular RNA species together with respective modifying enzymes highlighted.
  2. The RNA modifications are thoroughly described. While reading this review, I wondered if something is also known about the protein modification (PTMs) of mitoribosome components? The one or two sentences of comment could provide some balance and possibly highlight the missing research area.
  3. Section 2.3.2. of pseudouridine synthases is somewhat sloppy compared to other parts of the manuscript that were very precisely and carefully written. It would be great if authors can clearly discriminate between the actual pseudouridine synthases and the factors shown to associate with them in the so-called “16S RNA regulatory module” (Arroyo et al., 2016). To my knowledge, it is yet unclear how and if FASTKD2, NGRN and WBSCR16 mediate the RNA modification or rather bind and stabilize the 16S rRNA via a distinct mechanism (Antonicka et al., 2017). Also, the title of this section and the respective panel in Figure 3 could be more accurate.
  4. I somewhat miss the few words of comments on the global distribution of pathogenic variants in the mitoribosomes. Are there any structural or functional hotspots of disease-associated mutations in the mitoribosomes, and if yes, what can be the meaning behind them? Are we able to draw some conclusions? Or contrary, we have too little evidence yet? Can authors speculate briefly on it? It would add more value to this part of the manuscript to be more than just a list of the disease-associated changes and their structural consequences.
  5. Line 786: “mt-ACP (NDUFAB)” should be corrected to “mt-ACP (NDUFAB1)”.
  6. Lines 399-411 and 786-795. The mt-ACP/NDUFAB1 is a genuinely multifaceted protein with broad moonlighting functions in the mitochondria. Besides participating in the mt-LSU maturation, it is also an essential Complex I subunit and an important player in iron-sulfur cluster biogenesis and fatty acid synthesis, revealing a potent regulatory function of ACP interconnecting many different aspects of mitochondrial physiology. This feature is somehow missing (particularly in section 3.3.6.), as it influences the way how the disease-associated variants of mt-ACP should be interpreted.

Author Response

We wish to thank the referee for their time to assess our manuscript and their positive comments. We are also grateful to the referee for bringing up a number of specific pertinent points, which, as detailed in our responses below. It has been very helpful for revising the manuscript.

A few minor aspects might be considered:

  1. Subchapter 2.3. is very complex, and it would highly benefit from the additional figure that could help navigate through the text. For example, schematic representation of the modified areas of particular RNA species together with respective modifying enzymes highlighted.

We agree with the reviewer that posttranscriptional modifications of rRNA is a complex topic, and we have recently covered it in a separate review [Lopez Sanchez, M.I.G.; Cipullo, M.; Gopalakrishna, S.; Khawaja, A.; Rorbach, J. Methylation of Ribosomal RNA: A Mitochondrial Perspective. Frontiers in Genetics 2020, 11.], where we provided a figure (Fig 1) that shows rRNA modifications sites on mitoribosome. For this reason in this review, we only provide a very general description and refer to our previous work:

“As this topic was covered in detail by our recent review [60], we will focus on the main aspects here.”

2. The RNA modifications are thoroughly described. While reading this review, I wondered if something is also known about the protein modification (PTMs) of mitoribosome components? The one or two sentences of comment could provide some balance and possibly highlight the missing research area.

Thank you for highlighting this very interesting yet mainly unexplored research field. We added a short chapter summarizing the recent knowledge on the role of PTMs in ribosome biogenesis and highlighted the gaps of knowledge in regards to the mitoribosome (lines 441-475).

3. Section 2.3.2. of pseudouridine synthases is somewhat sloppy compared to other parts of the manuscript that were very precisely and carefully written. It would be great if authors can clearly discriminate between the actual pseudouridine synthases and the factors shown to associate with them in the so-called “16S RNA regulatory module” (Arroyo et al., 2016). To my knowledge, it is yet unclear how and if FASTKD2, NGRN and WBSCR16 mediate the RNA modification or rather bind and stabilize the 16S rRNA via a distinct mechanism (Antonicka et al., 2017). Also, the title of this section and the respective panel in Figure 3 could be more accurate.

We agree that this paragraph needed additional information for clarity. We adjusted the title (line 343) and added more information in the figure legend 3 (lines 181-185) and main text (lines 355-363) as suggested by this reviewer.

4. I somewhat miss the few words of comments on the global distribution of pathogenic variants in the mitoribosomes. Are there any structural or functional hotspots of disease-associated mutations in the mitoribosomes, and if yes, what can be the meaning behind them? Are we able to draw some conclusions? Or contrary, we have too little evidence yet? Can authors speculate briefly on it? It would add more value to this part of the manuscript to be more than just a list of the disease-associated changes and their structural consequences.

We thank the reviewer for this important observation. Initially, we mapped the pathogenic variants to  mitoribosome structures with the aim to categorize them based on distribution, but could not find any clear patterns. Instead, we provided a list of pathogenic variants and their structural consequences. As more mutations are being identified, we hope that future research may elucidate potential structural or functional hotspots for pathogenic variants but at this stage we do not want to yet speculate on this issue.

5. Line 786: “mt-ACP (NDUFAB)” should be corrected to “mt-ACP (NDUFAB1)”.

We have now corrected this. 

  1. Lines 399-411 and 786-795. The mt-ACP/NDUFAB1 is a genuinely multifaceted protein with broad moonlighting functions in the mitochondria. Besides participating in the mt-LSU maturation, it is also an essential Complex I subunit and an important player in iron-sulfur cluster biogenesis and fatty acid synthesis, revealing a potent regulatory function of ACP interconnecting many different aspects of mitochondrial physiology. This feature is somehow missing (particularly in section 3.3.6.), as it influences the way how the disease-associated variants of mt-ACP should be interpreted.

We thank the reviewer for their useful feedback. We have now addressed this point by adding the following: 

Lines 423-427:

“mt-ACP, also known as NDUFAB1, is a multifaceted protein with important functions in mitochondria (PMID: 31473256). Apart from being involved in mitoribosome biogenesis as part of the MALSU1–L0R8F8–mt-ACP module, it is involved in fatty acid synthesis, iron-sulfur cluster biogenesis, and is an integral component of respiratory complex I (PMID: 27540631; PMID: 30118679, PMID: 25209663).”

And lines 853-855: “Given the important, interconnected roles mt-ACP plays in mitochondrial metabolism [PMID: 31473256], additional studies are needed to elucidate the specific mechanisms linking genetic variants in NDUFAB1 and disease.”